# Integrated Management Practices for Canopy–Topsoil Improves the Grain Yield of Maize with High Planting Density

**DOI:** 10.3390/plants12102000

**Published:** 2023-05-16

**Authors:** Xuefang Sun, Xuejie Li, Wen Jiang, Ming Zhao, Zhuohan Gao, Junzhu Ge, Qing Sun, Zaisong Ding, Baoyuan Zhou

**Affiliations:** 1Shandong Provincial Key Laboratory of Dryland Farming Technology, College of Agronomy, Qingdao Agricultural University, Qingdao 266109, China; sunxuefang@qau.edu.cn (X.S.); lixuejie1126@163.com (X.L.); jwen1018@163.com (W.J.); sunqing0208@126.com (Q.S.); 2Institute of Crop Sciences, Chinese Academy of Agricultural Sciences/Key Laboratory of Crop Physiology and Ecology, Ministry of Agriculture and Rural Affairs, Beijing 100081, China; zhaoming@caas.cn (M.Z.); gaozhuohan@caas.cn (Z.G.); 3College of Agronomy, Resources and Environment, Tianjin Agricultural University, Tianjin 300392, China; gejunzhu@tjau.edu.cn

**Keywords:** maize, strip deep rotary, staggered planting, root, canopy, yield

## Abstract

Inappropriate spatial distribution of canopy and roots limits further improvements to the grain yield of maize with increased planting density. We explored an integrated management practice called strip deep rotary with staggered planting (SRS) which includes comprehensive technology for both canopy layers and topsoil. Here, field experiments were conducted under two maize cropping systems (spring maize and summer maize) to evaluate the effect of SRS on the spatial distribution of the canopy and roots for maize under high planting density (90,000 plants ha^−1^) and to determine the physiological factors involved in yield formation. Compared with conventional management practices (no-tillage with single planting, NTS), SRS decreased the LAI of the middle to top layers while improving the light distribution of the middle and lower layers by 72.99% and 84.78%, respectively. Meanwhile, SRS increased the root dry weight density and root sap bleeding by 51.26% and 21.77%, respectively, due to the reduction in soil bulk density by an average of 5.08% in the 0–40 cm soil layer. SRS improved the SPAD in the ear and lower leaves and maximized the LAD, which was conducive to dry matter accumulation (DMA), increasing it by 14.02–24.16% compared to that of NTS. As a result, SRS increased maize grain yield by 6.71–25.44%. These results suggest that strip deep rotary combined with staggered planting noticeably optimized the distribution of light in the canopy and reduced the soil bulk density to promote root vitality and growth, to maintain canopy longevity, and to promote the accumulation of dry matter, which eventually increased the grain yield of the maize under high planting density conditions. Therefore, SRS can be considered a better choice for the sustainable high yield of maize under high-density planting conditions in the NCP and similar areas throughout the world.

## 1. Introduction

In order to meet the global food demand, crop yields need to increase by 70–100% by 2050, with an average annual increase of 44 million tons [1,2,3]. Maize (*Zea mays* L.), as one of the most important food crops, has an important role in expanding overall grain production capacity. However, owing to the limitation of arable land area, increasing the grain yield per unit area is the key way to ensure food security in China. Therefore, great effort should be made to improve management practices to further increase maize yield per unit area.

Increasing planting density is considered to be an effective way to further increase maize grain yield [4]. However, when the planting density is too high, the competition among individual plants intensifies, and the available resources (such as light, heat, water, fertilizer, etc.) for each plant are limited [5,6]. It is difficult to achieve high yield with high planting density [7,8]. With increasing planting density, the light transmittance of the canopy is poor, especially with regard to the reduction in light transmittance to the lower leaves [9], which causes the leaves of the middle and lower layers to be under low light intensity levels and to be lower than the compensation point. Thus, the photosynthetic potential of the shaded leaves is inhibited [10,11]. The distribution ratio of maize photosynthetic products to the root system is reduced under weak light stress, resulting in a significant decrease in root biomass, root length density, and root surface area, and a weakening of root vitality [12,13,14]. Therefore, a reasonable planting method is necessary to optimize the canopy structure and the distribution of canopy light.

Long-term minimal tillage or no tillage during the maize growing season increases the soil bulk density, which restricts the water and nutrient distribution as well as the root development and extension throughout the soil profile [15,16]. As a result, further increasing in the gain yield of maize under high plant density conditions is seriously limited [17]. Therefore, appropriate tillage practices are also necessary for improving maize production under high plant density conditions. One of these practices is subsoiling, which can improve the physical characteristics of soil by breaking the plow pan, enhancing the soil permeability, soil water storage, and moisture conservation capacity [18,19,20,21], and which can also optimize root distribution [22].

Many researchers have demonstrated that greater integration of agronomic management factors, such as tillage, planting method, nutrients, and water management practices, is an effective way to meet the requirements of crop production [23,24,25,26,27,28]. We explored an integrated management practice, named strip deep rotary with staggered planting (SRS), which could significantly improve the grain yield of maize with high planting density. However, further studies are needed to evaluate the effect of SRS on the spatial distribution of the canopy and the roots of maize under high planting density conditions and to determine the physiological factors involved in yield formation. Therefore, in this study, we aimed to evaluate the effects of SRS on (i) canopy structure (LAI), light distribution, and photosynthetic index (DMA, LAD, MNAR, and SPAD); (ii) soil bulk density, root dry weight density, and root activity; and (iii) the correlation between yield and canopy and tillage layer indexes. The results of these investigations should provide theoretical support for determining the physiological processes of yield formation under the condition of synergetic cultivation and canopy factors and eventually provide guidance for sustainable maize production in the North China Plain (NCP).

## 2. Results

### 2.1. Canopy Structure and Light Distribution

The leaf area index (LAI) reached the maximum value at the anthesis stage (R1), and the maximum LAI (LAImax) values ranged from 5.17 to 7.18 in different cropping systems in 2015–2016 (Figure 1). With the exception of the LAImax value in the summer maize in 2015, which was 11.98% higher than the NTS, there were no other significant differences between the SRS and the NTS. Compared to the NTS, the LAI of the SRS maintained higher values at the post-anthesis stage. At the R3 stage, the LAI of the SRS was 16.72% (spring maize) and 18.84% (summer maize) in 2015 and 11.98% (spring maize) and 10.86% (summer maize) in 2016, higher than the NTS. At maturity (R6) stage, the LAI of the SRS increased by 76.52% (spring maize) and 28.84% (summer maize) in 2015 and 20.19% (spring maize) and 42.05% (summer maize) in 2016. From the R1 stage to the R6 stage, the LAI reduction of SRS was 69.06% and 37.39% (spring maize) and 47.50% and 47.13% (summer maize) in 2015–2016; however, the NTS decreased by 82.12% and 46.08% (spring maize) and 54.37% and 61.46% (summer maize) in 2015–2016.

The LAI distribution of the canopy, from top to bottom, was as follows (Figure 2A,B): Compared with the NTS, the LAI from bottom to ear significantly increased by 11.85% (in spring maize) and 19.54% (in summer maize) under the SRS treatment. However, the SRS significantly reduced the LAI from ear to top by 14.54% and 16.83% in spring and summer maize, respectively. At the top layer, the LAI was SRS < NTS.

The canopy light transmittance in different layers (top, middle, and bottom) at the anthesis stage under the NTS and SRS are shown in Figure 2C,D. The canopy light transmittance showed a decreasing trend from the top leaf layer to the bottom leaf layer. The SRS treatment showed higher light transmittance between the spring maize and the summer maize. The light distribution of the middle layers and bottom layers under the SRS was increased by 19.99% and 243.04%, respectively, in spring maize and 70.42% and 35.29%, respectively, in summer maize. The top leaf layer of the SRS treatment was 4.49% and 6.35% higher than the NTS in spring maize and summer maize, but there was no obvious difference between the NTS and the SRS in the top leaf layer.

### 2.2. SPAD of Canopy Leaves

To better evaluate the effect of strip deep rotary combined with staggered planting on leaf senescence, we measured the SPAD values of the top, ear, and lower leaves of spring and summer maize at the anthesis stages in 2015 and 2016 (Figure 3). The SRS treatment displayed higher SPAD values. The SPAD under the NTS was 4.89% to 8.34% (ear leaves) and 6.91% to 9.52% (lower leaves) lower than that in the SRS treatment. However, the SPAD of the top leaves between the two treatments was not significantly different.

### 2.3. MLAI, LAD, and MNAR

In spring maize, compared to the NTS treatment, MLAI and LAD under the SRS was slightly increased by 10.89–13.40% in 2015 and 2016 (Figure 4). The SRS treatment in 2016 caused a significantly higher mean net assimilation rate (MNAR) compared with the NTS (an increase of 9.45%). However, the MNAR in 2015 was reduced by 9.80% in the SRS treatment.

In summer maize, a higher MLAI and LAD were found under the SRS treatment, with increasing values of 9.90–11.87% in 2015 and 2016. Over the two years, the MNAR was SRS > NTS, and the increases in SRS were 7.06% and 10.96%, respectively.

### 2.4. Soil Bulk Density and Root Bleeding Sap Content

The soil bulk density showed an increasing trend with the deepening of the plough layer (Figure 5A). The effect of the SRS on the characteristics of the plough layer was mainly to reduce the bulk density in each soil layer. At the 0–10, 10–20, 20–30, and 30–40 cm soil layers, the SRS treatment decreased by 7.85%, 3.83%, 5.52%, 4.91%, and 3.77%, respectively, compared with the NTS. To a certain extent, the root bleeding sap content is an important indicator for qualitative analysis of root activity (Figure 5B). The sap content in roots under the SRS was higher by 21.77%.

### 2.5. Root Dry Weight Density

The dry weight of the roots decreased significantly with the increase in soil depth (Figure 6). The root dry weight density in the soil layer at 0–50 cm was 0.13 kg m^–3^ for the NTS and 0.19 kg m^–3^ for the SRS. Compared to the NTS, the root dry weight was increased by 33.26% (0–10 cm), 6.92% (10–20 cm), 30.82% (20–30 cm), −16.84% (30–40 cm), and 1.81% (40–50 cm).

### 2.6. Dry Matter Accumulation and Change Rate

Maize DMA differed by management practice, cropping system, and year (Figure 7). The total DMA of spring maize was 10.95% (2015) and 26.11% (2016) higher than summer maize. On average, the DMA of SRS was 26.24 Mg ha^−1^ (range: 22.37–28.61 Mg ha^−1^) and the NTS was 21.94 Mg ha^−1^ (range: 18.01–22.74 Mg ha^−1^). The total DMA under the SRS was 14.02% and 17.66% higher than the NTS in 2015 and 24.07% and 24.16% higher in 2016. At the anthesis stage, the difference in DMA between SRS and NTS was not significant, even though it exceeded that of the spring maize in 2016. However, at the post–anthesis stages, the DMA under the SRS was increased by 19.77% and 26.64% in spring maize and 28.95% and 46.78% in summer maize. It can be seen that the positive regulatory effects of SRS treatment on DMA were greater in summer than in spring.

The DMA dynamics followed the logistic equation, and the determination coefficient R^2^ of the fitting equation was above 0.9869, which was extremely significant (at the 0.01 level). The DMA rate was affected by the management practice and sowing season (Table 1). Under the SRS, the Wmax, Gmax, and Gmean were, on average, 19.15%, 24.93%, and 24.89% higher than under the NTS in spring maize. In summer maize, significantly greater values for Tmax, Wmax, Gmax, and D were found under the SRS compared with the NTS.

### 2.7. Yield and Yield Components

Yield varied considerably, with significant differences between the two management practices and the different cropping systems in 2015 and 2016 (Table 2). Yields ranged from 7.6 to 11.9 t ha^−1^. In the two cropping systems, the yield of the SRS increased by 7.41% and 6.71% in 2015 and by 25.44% and 14.91% in 2016 compared with the NTS. We found no obvious differences in yield between spring maize and summer maize. However, there was a significant difference in maize yield between the two years, and the yield in 2015 was 32.11% higher than in 2016. The results of the interactive analysis showed that the yield was only significantly affected by the year × management practices (*p* < 0.01).

The number of harvested ears under the SRS was 23.47% higher than the NTS in spring maize of 2016, but in another treatment, there was no significant difference between the SRS and the NTS. The grain number per ear was higher under the SRS and was increased by 1.78% and 3.43% in 2015 and 7.63% and 7.41% in 2016. Under the SRS, the 1000-grain weight was increased by 4.04% and 4.39% in 2015 and by 6.47% in summer maize in 2016.

### 2.8. Correlation Analysis

According to the correlation analysis (Figure 8), the yield was positively correlated with the aboveground dry matter accumulation (DMA) (R^2^ = 0.627 **). In addition, the higher DMA should theoretically benefit the MLAI, LAD, T-LT, M-LT, B-LT, T-SPAD, E-SPAD, L-SPAD, RDW, and RBS. The DMA and SBD displayed an extremely significant negative correlation. The MLAI and LAD were significantly, positively correlated with T-SPAD, E-SPAD, L-SPAD, RDW, and RBS. The root dry weight density and root bleeding sap content had negative correlations with soil bulk density, and the correlation coefficients were −0.962 ** and −0.996 **.

## 3. Discussion

Modifying canopy architecture is an important way to achieve high yield for maize under high-density conditions [29]. Leaves are the main organs for photosynthesis and carbohydrate accumulation [30]. Leaf area index (LAI) is a key parameter for characterizing the structure and function of the canopy [31]. Previous studies also reported that high-yield maize populations have a higher leaf area index [32,33]. In this study, we found that the MLAI under the treatment of SRS was 10.89–13.40% higher than the NTS, and the reason was mainly that SRS could keep the LAI higher at post-anthesis, but there was no significant difference in LAImax at anthesis between the SRS and NTS (Figure 1). This indicated that the SRS treatment could delay the senescence of leaves after anthesis and maintain a higher photosynthetic function in the leaves. Previous studies reported that higher leaf area after anthesis was an important guarantee for maintaining dry matter accumulation [34], and early senescence of leaves during the grain filling stage was not beneficial to grain yield improvement [35]. The findings supported our results. LAI is an important index reflecting plant growth and development and the utilization of light energy, and it is also the main factor affecting the dry matter accumulation and the formation of yield [36,37]. An optimized canopy structure allows light to be reasonably, vertically distributed to each leaf layer, which enables the leaves in the lower canopy to obtain more light energy and therefore, ensures that the lower leaves are in good light conditions, which allows the maize to achieve a higher grain yield [11,38]. In our research, the light transmittance at the upper canopy was not different between SRS and NTS; however, the light transmittance of the middle and lower canopy of the SRS was higher than that of the NTS (Figure 2). The light distribution of the middle and lower layers was improved by 72.99% and 84.78%, respectively. The SRS decreased the LAI of the middle to top layers and this ensured that more light was distributed to the middle and lower layers of the canopy. At the same time, the leaf area index below the ear position leaves showed SRS > NTS, and the higher LAI could also increase the interception of light energy in the middle and lower parts of the canopy to maintain the leaf photosynthetic function in the lower leaf layer. The SRS with small, double-row, staggered planting could effectively expand the plant’s individual ecological niche in the canopy, relieve the pressure of mutual shading between individuals, and increase the light distribution in the middle and lower parts of the canopy. This maximized light absorption and delayed the senescence of the middle and lower leaves.

An appropriate increase in the light interception rate could improve photosynthetic capacity [39], and optimizing the photosynthetic production capacity of the canopy and preventing the premature senescence of leaves are the keys to tapping the potential yield of maize. Chlorophyll content is closely related to the photosynthetic performance of leaves, and the SPAD value is often used to represent the relative content of chlorophyll in active leaves [40]. Under the SRS treatment, the SPAD at the ear and lower leaves were, on average, 7.12% and 8.97% higher than that of NTS (Figure 3). Wu et al. pointed out that improving the planting method to optimize the canopy structure could improve the decay rate of the SPAD value in the middle and lower leaves after anthesis, delay senescence, and ensure the supply capacity of the assimilation “source” during grain filling [41]. Our study also showed that there was a very significant positive correlation between the yield and the light interception rates of the middle and lower canopies, and the correlation coefficients were 0.918 ** and 0.745 ** (*p* < 0.01) (Figure 8). This was consistent with our research results. This suggested that the SRS was conducive to the establishment and photosynthetic function of leaves in the middle and lower canopy by expanding the light-receiving area of the population to strengthen the “source” supply capacity in the late growth period, thereby increasing the yield. LAD is a physiological index to characterize the green leaf area of the crop population and it is determined by leaf area and longevity, which play important roles in regulating the photosynthetic activity of maize [32]. Our results showed that the LAD under the SRS was higher than under the NTS, and this indicated that the SRS could maintain higher leaf area retention. This was consistent with the result that the SRS treatment could delay the senescence of leaf area and maintain the photosynthetic function of the leaves at post-anthesis. The net assimilation rate (MNAR) is one of the important indicators by which the photosynthetic capacity of crops is measured. In this study, it was found that the MNAR of spring maize in 2015 was SRS < NTS, and the other three stages showed that SRS treatment had a higher MNAR. This might be because the spring maize in 2015 had a higher LAD under SRS treatment. Due to the dilution effect, the dry matter accumulation had decreased per unit of area and per unit of time.

The impact of tillage systems on soil structure plays a major role in crop productivity [42,43]. Previous reports demonstrated that shallow plowing and rotary tillage mainly disturb soil at the 0–15 cm depth. A series of problems, such as a shallow tillage layer, a thick plow bottom, and subsoil compaction, were the result of long–term rotary tillage, which inhibited the increase in crop yield [44,45]. Subsoiling, a type of conservation tillage, can loosen the soil structure without destroying soil layers, as well as increase soil porosity, air permeability, and soil water storage, in order to promote root growth and nutrient distribution into deeper soil layers, which enhances crop growth and increases yield [46,47]. Our results indicated that the SRS decreased the soil bulk density (SBD) in the 0–40 cm soil layer to 6.47% lower than the NTS. This showed that the SRS treatment effectively reduced the soil bulk density and broke the hard plough bottom under our study conditions. Root distribution significantly influences the uptake of nutrients and soil moisture [48]. Piao et al. reported that the roots in the subsoil tillage treatments tended to converge around the plant centers and were mainly distributed in the 0–20 cm soil layer [49]. In our study, the SRS treatment increased the root dry weight density (RDW) at 0–30 cm, and the rate increases were 33.26%, 6.92%, and 30.82%. Because the maize plants under the SRS treatment were planted in a staggered pattern, compared with single–row planting, the spatial distance between the roots was larger. Therefore, by changing the spatial niche of the roots, the crowding of the roots in the surface soil was reduced. At the same time, under the effect of strip deep loosening, the growth of the roots in the lower layer of the soil was promoted. Root bleeding sap is an indicator of root pressure, plant growth potential, and root activity [50]. Stronger root activity is conducive to the absorption of water and nutrients by the root system and promotes the growth of aboveground plants [51]. We found that, under the SRS treatment, the root bleeding sap significantly increased by 21.77% compared to that of the NTS treatment. This suggested that SRS could improve the vitality of the root system and delay root senescence. These results were consistent with those of our previous research [15]. Thus, SRS mainly disrupted the plough pan and created a more stable root environment to enhance the stress resistance of maize and allow an extension of the root system into deeper soil depths, resulting in higher root activity.

Dry matter accumulation is the basis for maize grain yield formation [52]. The establishment of vegetative organs, such as stems and leaves, before anthesis provides the basis for yield, and the accumulation of photosynthetic metabolites at post–anthesis is the key to yield formation [15,53]. Previous studies reported that the optimization of light distribution in the canopy could maintain high photosynthetic capacity in leaves at later growth stages, which is helpful to the final accumulation of aboveground biomass [54,55]. Liu et al. pointed out that further increases in grain yield should be dependent on increasing dry matter to better match light availability [56]. Sun et al. also showed that subsoiling could improve the dry matter accumulation above ground [15]. In this study, the total DAM of the SRS was higher (range: 14.02–24.16%) than that of the NTS, and the difference in dry matter between the two treatments was mainly at the post–anthesis growth stage (Figure 7). Further analysis of dry matter accumulation parameters (Table 1) showed that the population had a higher Gmean under the SRS treatment in spring maize. Compared with NTS, it increased by 28.0% and 16.67%; however, in summer maize, the main reason for more dry matter accumulation under the SRS treatment was that it had a longer effective accumulation period (D). The D of the SRS was, on average, 58.82% higher than the D of the NTS. Thus, the higher average accumulation rate and prolongation of the growth duration was attributed to the increase in dry matter accumulation under SRS. The correlation analysis showed that there was a very significant negative correlation between soil bulk density and dry matter accumulation above ground, and a significant positive correlation between root activity, root dry weight density, MLAI, LAD, T–LT, M–LT, B–LT, T–SPAD, E–SPAD, L–SPAD, and DMA. These phenomena indicated that the SRS treatment could optimize the structure of the tillage layer and the canopy simultaneously to enhance the photosynthetic carbon assimilation capacity of the aboveground canopy as well as dry matter accumulation. The yield under the SRS treatment was increased by 6.71–25.44% compared to the NTS. The SRS had a greater capacity of DMA and the correlation analysis showed that grain yield was positively correlated with the aboveground DMA. Our results were consistent with previous studies [57,58].

## 4. Materials and Methods

### 4.1. Experimental Site

Field experiments under two maize cropping systems (spring maize and summer maize) were conducted in 2015 and 2016 at the Langfang experimental station of the Chinese Academy of Agricultural Sciences (35°11′30″ N, 113°48′ E), Hebei province. The soil of the region is sandy loam. The total organic matter, alkaline N, available P, available K, and PH were 13.27 g kg^−1^, 59.15 mg kg^−1^, 47.80 mg kg^−1^, 163.43 mg kg^−1^, and 8.36, respectively (in spring maize); and 16.59 g kg^−1^, 80.67 mg kg^−1^, 25.66 mg kg^−1^, 136.10 mg kg^−1^, and 8.5, respectively (in summer maize), in the 0–20 cm depth layer prior to maize planting in 2015.

### 4.2. Experimental Design and Field Management

Experiments were arranged in a randomized complete block design with three replicates, at a plant density of 90,000 plants ha^−1^, under both a spring maize cropping system (sowing in mid to late May every year) and a summer maize cropping system (sowing after winter wheat is harvested in mid to late June every year). Both of these two cropping systems included two management treatments (SRS and NTS) (Figure 9): SRS, integrated management practices for canopy–topsoil treatment with strip subsoiling in the tillage layer and small double–row, staggered planting technology in the canopy layer; and NTS, for which we used the maize no–tillage sowing machine (Nonghaha Machinery Co., Ltd., Shijiazhuang City, Hebei, China) with seed planted in a single row. In each experiment and year, we applied 225 kg N ha^−1^, 120 kg P_2_O_5_ ha^−1^, and 150 kg K_2_O ha^−1^ as basal fertilizer and an additional N fertilizer (135 kg N ha^−1^) at the twelfth leaf stage. The amount of fertilizer applied was based on the existing levels of N, P, and K, as determined from soil tests, to ensure that there were no nutrient deficiencies. The maize hybrid Zhongdan 909 (ZD909) was selected as the experimental material for testing, which was provided by the Institute of Crop Science, Chinese Academy of Agricultural Sciences. Each experiment plot (50 m × 7.2 m) consisted of twelve rows with three replications. Fertilizer and water management were optimized throughout the maize growing season. Weeds, pests, or diseases were well controlled in any of the plots. The specific planting time and the period corresponding to each growth period of maize are shown in Table 3 and the inter-annual climatic conditions are shown in Figure 10.

### 4.3. Sampling and Measurement Method

#### 4.3.1. Light Transmittance at Different Canopy Layers

At the anthesis stage, we used a canopy analyzer (AccuPAR LP-80) to measure the light transmittance of three different leaf layers. The measuring position was between two rows of maize. The canopy was divided into three leaf layers (top, middle, and bottom). The top layer is defined as the position of the converse third leaf to the top of the plant; the middle leaf layer is defined as the position of the ear leaf; and the bottom leaf is defined as the near-ground leaf.

#### 4.3.2. Leaf Area Index (LAI)

At V6, V12, anthesis, R3, and R6 stages, the lengths and maximum widths of each green leaf of three plants with the same growth vigor for each plot were measured. All the expanded and unexpanded leaves were recorded and the leaf area was computed based on the following formula [59]:Leaf area = length × width × 0.75 (fully expanded leaf)
Leaf area = length × width × 0.5 (incompletely expanded leaf)

LAI was calculated as follows:LAI = leaf area (m^2^ plant^−1^) × plant density plants ha^−1^/10,000 m^2^ ha^−1^

#### 4.3.3. Mean LAI (MLAI), Mean Net Assimilation Rate (MNAR), and Leaf Area Duration (LAD)

According to the calculation method of parameters in the quantitative expression of production performance [60], we adopted the method of normalization to calculate the mean LAI (MLAI). MNAR was determined according to the method of Hou et al. [61]. Leaf area duration (LAD) is the accumulation of photosynthetic leaf area per unit of land area during the growth period, and it is defined as:LAD = MLAI × D
where D is the days in the whole growth period.

#### 4.3.4. Dry Matter Accumulation (DMA)

Three plants in each plot were collected to determine dry matter accumulation (DMA) at V6, V12, anthesis, R3, and R6. The dry matter was dried at 105 °C for 30 min and then at 75 °C to constant water content before being weighed. DMA conformed to a logistic equation (*y* = *a*/(1 + *be^−cx^*)) and the daily DMA rate was calculated as the first-order derivative of the following logistic equation [62]:$$\frac{dy}{dx}=\frac{abce^{-cx}}{{\left(1+be^{-cx}\right)}^{2}}$$
where *a* is the final DMA, *b* is the initial DMA, *c* is the parameter representing growth rate, *d* is derivation operation, *e* is a natural constant, *x* is the days for DMA, and *y* is the weight of DMA.

DMA parameters were calculated according to the following equations: Required days for the DMA rate to reach the maximum (Tmax, d): Tmax = ln*b*/*c*; The weight of dry matter accumulation at the maximum DMA rate (Wmax, g): Wmax = *a*/[1 + *b* × exp(−*c* × Tmax)]; Maximum accumulation rate of dry matter (Gmax, kg ha^–1^ d^–1^): Gmax = *a* × *b* × *c* × exp(−*c* × Tmax)/[1 + *b* × exp (−*c* × Tmax)]^2^; Average DMA rate (Gmean, kg ha^−1^ d^−1^): Gmean = *ac*/6; Effective accumulation duration of dry matter (D, d): D = *a*/Gmean.

#### 4.3.5. SPAD Values

SPAD values of the upper leaf (the converse third leaf), the ear leaf, and the lower leaf (the fourth leaf below the ear leaf) were determined using a SPAD-502 chlorophyll meter at anthesis. Three points were measured for each leaf and three plants were measured for each plot.

#### 4.3.6. Root Dry Weight and Soil Bulk Density

At anthesis, the root samples were collected according to Wang’s methods [63]. The plant was in the center of the soil samples, with line spacing and planting spacing of 60 cm (the distance from the left and right sides of the plant and the front and back sides of the plant were all 30 cm) and a depth of 0–50 cm. All the roots in each soil block were harvested and dried to weight.

Soil bulk density was measured using the cutting ring method at the anthesis stage. The soil samples from depths of 0–10, 10–20, 20–30, 30–40, 40–50 cm soil layers in each plot were collected for measuring soil bulk density by using a 100 cm^3^ standard cutting ring. The soil samples were dried to constant weight at 105 °C to determine the bulk density.

#### 4.3.7. Root Bleeding Sap

Root bleeding sap collection at anthesis was carried out according to Sun’s methods [15]. In each plot, three plants with the same growth were cut manually at 10 cm above the ground, removing the aboveground parts above 10 cm to determine the root bleeding sap amount. The diameter of the stem was measured using a vernier caliper to calculate the cross-sectional area of the stem. The weighed absorbent cotton (W0) was held tightly against the stem incision and tied with a plastic bag. Root bleeding sap was collected from 6:00 pm to 6:00 am the next day. Then, we recorded the accurate time as t, took it off, and weighed it as W1 to calculate the root bleeding sap intensity.

### 4.4. Statistical Analysis

The data were prepared using Microsoft Excel 2010, and statistical analyses were performed using SPSS 19.0 software (SPSS lnc., Chicago, IL, USA). Means were compared using Duncan’s new multiple range method test at a probability (*p*) level of 0.05. A two-way analysis of variance (ANOVA) was conducted to compare the differences between the maize yields and DMA. All graphics were drawn using Sigma plot 12.5 (Systat Software, San Jose, CA, USA).

## 5. Conclusions

Integrated management practices for canopy–topsoil of strip deep rotary tillage combined with staggered planting could improve the structure of topsoil to provide a good environment for root growth. SRS increased the root dry weight density and root sap bleeding by 51.26% and 21.77%, by decreasing soil bulk density across the 0–40 cm soil layers. Meanwhile, SRS could also optimize the distribution of canopy light by changing the spatial ecological niche of plants, expanding the spatial distance between plants and avoiding mutual shading. This promotes higher dry matter accumulation in the canopy and the maintenance of a higher leaf area index. The positive effect of SRS on yield was 6.71–25.44% under both cropping systems. Therefore, strip deep rotary tillage combined with staggered planting is a promising method by which to achieve sustainable maize yield under high-density planting conditions in the NCP and similar areas throughout the world.

## Figures and Tables

**Figure 1 plants-12-02000-f001:**
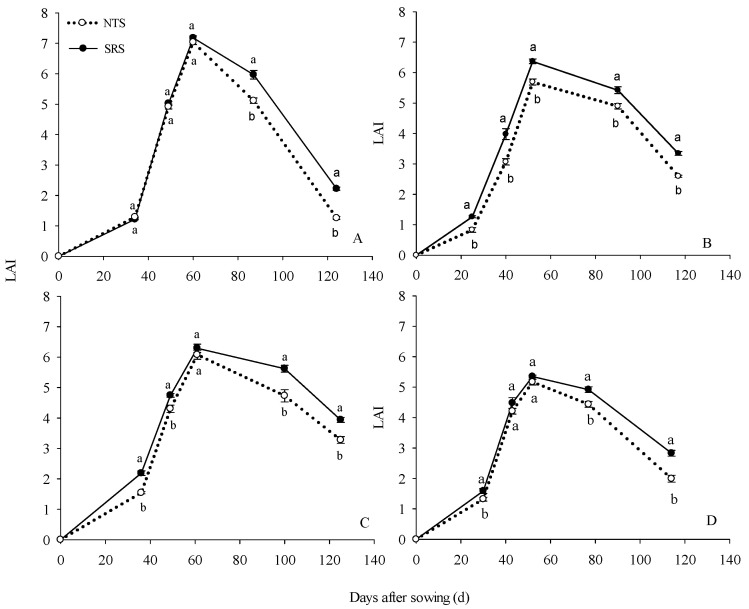
The maize leaf area index (LAI) under two cropping systems by different management practices in 2015 and 2016. NTS: no-till with single row planting. SRS: strip deep rotary combined with staggered planting. Values are mean ± SE (n = 3). Different lower-case letters adjacent to the error bars denote significant pairwise differences between means for a given date after sowing (*p* < 0.05; Duncan test). (**A**): LAI of spring maize in 2015; (**B**): LAI of summer maize in 2016; (**C**): LAI of spring maize in 2016; (**D**): LAI of summer maize in 2015.

**Figure 2 plants-12-02000-f002:**
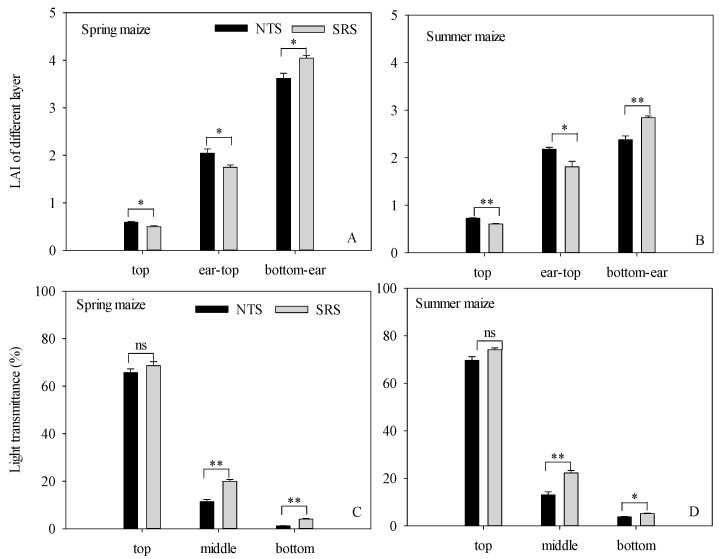
The LAI and light transmittance at different canopy layers in response to different management practices in spring maize and summer maize in 2016. (**A**): LAI at different canopy layers in spring maize; (**B**): LAI at different canopy layers in summer maize; (**C**): light transmittance in spring maize; (**D**): light transmittance in summer maize. NTS: no-tilling with single-row planting; SRS: strip deep rotary combined with staggered planting. *, *p* < 0.05; **, *p* < 0.01; ns, not significant.

**Figure 3 plants-12-02000-f003:**
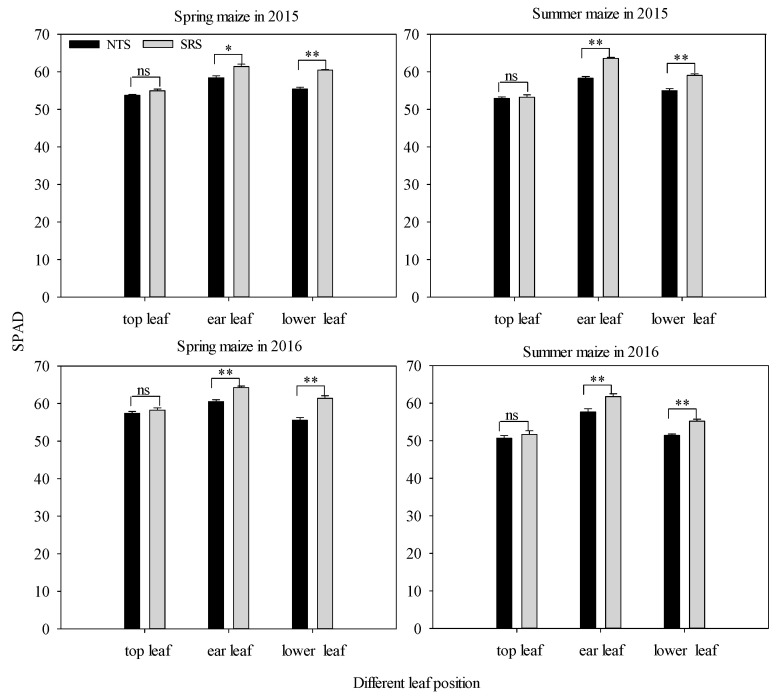
The values of SPAD at different leaf layers under two cropping systems in response to different treatments in 2015 and 2016. NTS: no-till with single row planting; SRS: strip deep rotary with staggered planting. Values are mean ± SE (n = 3). * indicates significance at *p* < 0.05, ** indicates significance at *p* < 0.01. ns indicates no significance, *p* > 0.05.

**Figure 4 plants-12-02000-f004:**
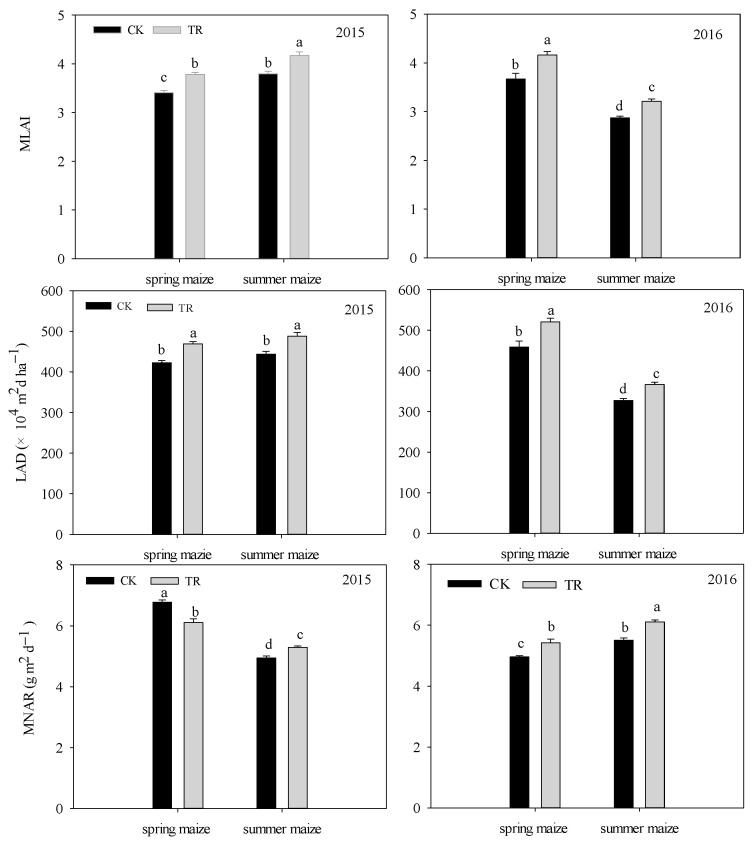
The maize average leaf area index (MLAI), LAD, and MNRA under two cropping systems in response to different treatments in 2015 and 2016. NTS: no-till with single row planting; SRS: strip deep rotary combined with staggered planting. Values are mean ± SE (n = 3). Different lower-case letters adjacent to the error bars denote significant pairwise differences between means for a given management practice (*p* < 0.05; Duncan test).

**Figure 5 plants-12-02000-f005:**
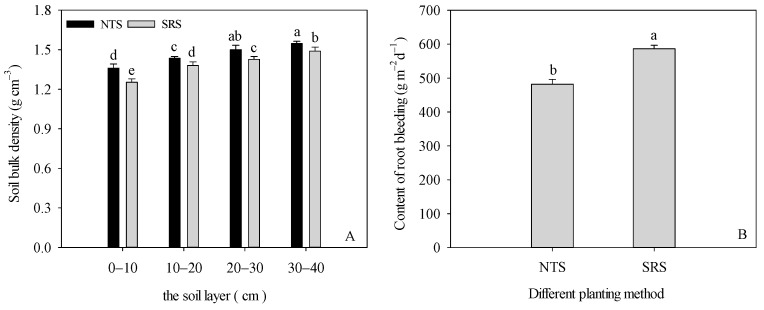
Effects of strip deep rotary with staggered planting on soil bulk density and root bleeding content. (**A**): soil bulk density; (**B**): root bleeding content; NTS: no-till with single row planting; SRS: strip deep rotary combined with staggered planting. Values are mean ± SE (n = 3). Different lower-case letters adjacent to the error bars denote significant pairwise differences between means for a given management practice (*p* < 0.05; Duncan test).

**Figure 6 plants-12-02000-f006:**
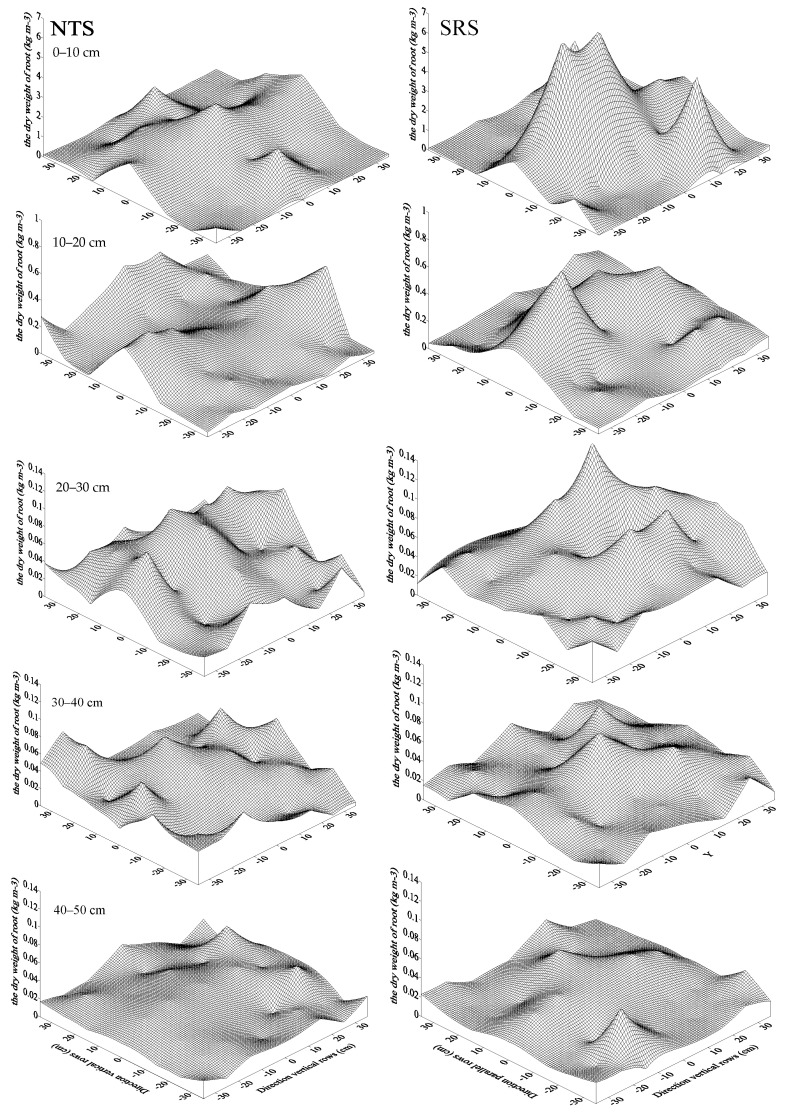
Effects of strip deep rotary with staggered planting on the root dry weight density. NTS: no–till with single row planting; SRS: strip deep rotary combined with staggered planting.

**Figure 7 plants-12-02000-f007:**
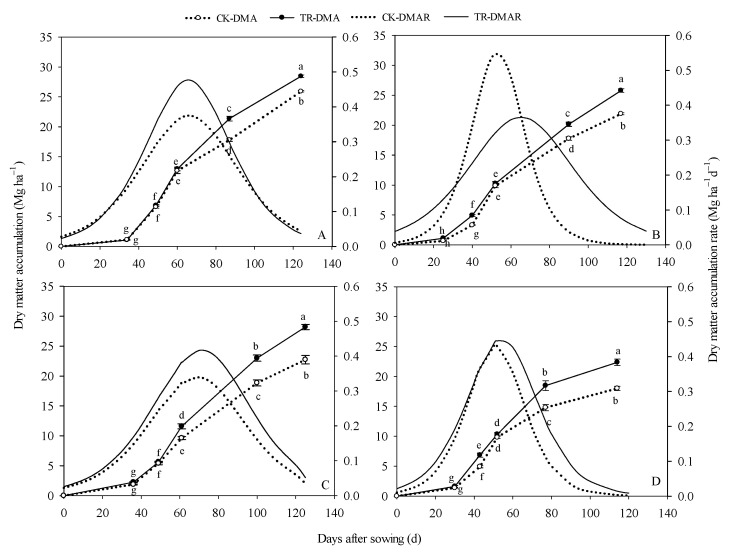
The dry matter accumulation (DMA) and dry matter accumulation rate (DMAR) of maize in response to different treatments in 2015 and 2016. (**A**): spring maize in 2015; (**B**): summer maize in 2015; (**C**): spring maize in 2016; (**D**): dry matter accumulation of summer maize in 2016. Values are mean ± SE (n = 3). Different lower-case letters adjacent to the error bars denote significant pairwise differences between means for a given date after sowing (*p* < 0.05; Duncan).

**Figure 8 plants-12-02000-f008:**
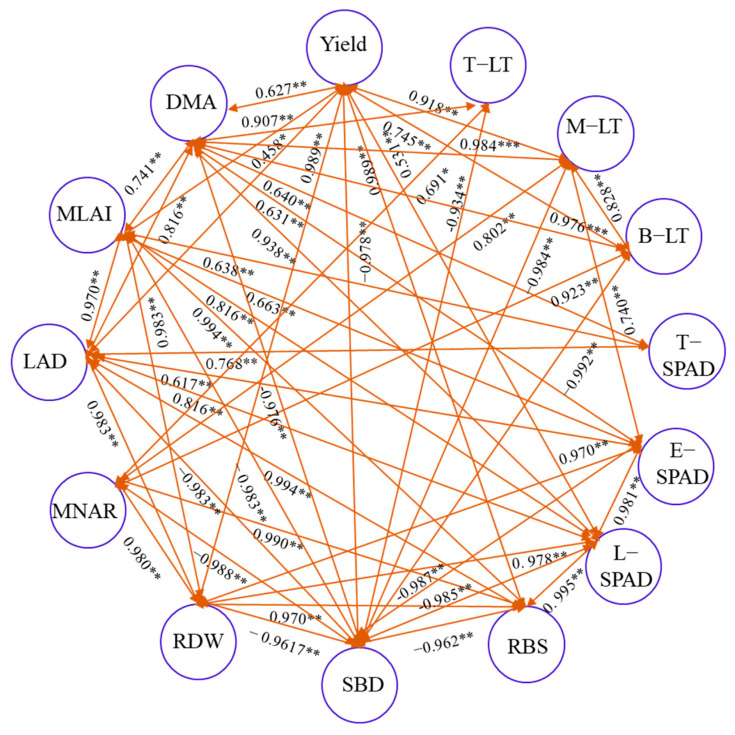
Correlation analysis of maize characteristics response to strip deep rotary combined with staggered planting management (*: *p* < 0.05; **: *p* < 0.01;***: *p* < 0.001). The DMA, MLAI, LAD, MNAR, RDW, SBD, and RBS are aboveground dry matter accumulation, mean leaf area index, leaf area duration, mean net assimilation rate, root dry weight density, soil bulk density, and root bleeding sap content. T-SPAD, E-SPAD, and L-SPAD are SPAD readings of top third leaf, ear leaf, and below fourth leaf from ear leaf. T-LT, M-LT, and B-TL are the light transmittance of the upper, middle, and bottom leaf layers.

**Figure 9 plants-12-02000-f009:**
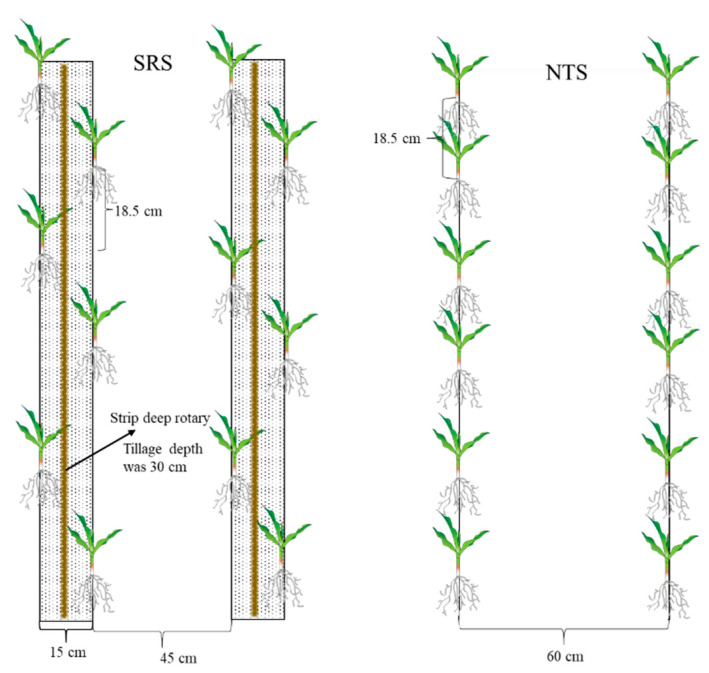
Schematic diagram of planting patterns for different cropping practices. SRS is integrated management practices for canopy–topsoil treatment with strip subsoiling in the tillage layer and small, double-row, staggered planting technology in the canopy layer; NTS is no-tillage with seed planted in a single row.

**Figure 10 plants-12-02000-f010:**
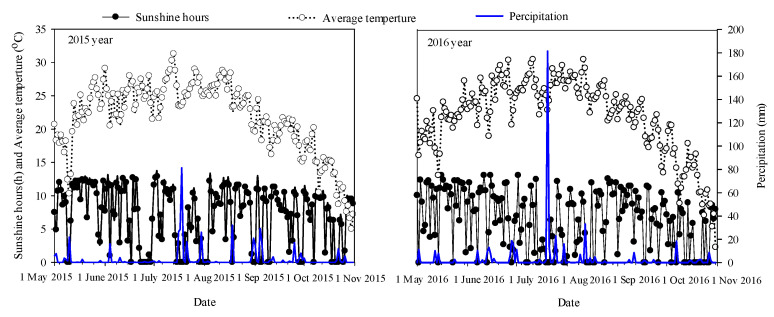
Daily temperature, sunshine hours, and precipitation during the maize growing seasons of 2015 and 2016.

**Table 1 plants-12-02000-t001:** The dry matter accumulation characteristic parameters of maize under two cropping systems in response to different management practices in 2015 and 2016.

Year	Cropping System	Treatment	Equation	R^2^	Tmax	Wmax	Gmax	Gmean	D
2015	spring maize	NTS	y = 25.123/(1 + 51.375e^(−0.060x)^)	0.9739 **	66.11 a	12.56 b	0.37 b	0.25 b	100.70 a
		SRS	y = 28.420/(1 + 81.514e^(−0.067x)^)	0.9839 **	65.73 a	14.21 a	0.48 a	0.32 a	89.61 b
	summer maize	NTS	y = 20.097/(1 + 353.611e^(−0.110x)^)	0.9817 **	53.17 b	10.05 b	0.56 a	0.37 a	54.36 b
		SRS	y = 26.366/(1 + 38.892e^(−0.057x)^)	0.9876 **	64.49 a	13.18a	0.37 b	0.25 b	105.69 a
2016	spring maize	NTS	y = 22.918/(1 + 56.721e^(−0.059x)^)	0.9918 **	68.97a	11.46 b	0.34 b	0.23 b	101.18 a
		SRS	y = 28.687/(1 + 63.129e^(−0.058x)^)	0.9930 **	71.30 a	14.34 a	0.42 a	0.28 a	103.20 a
	summer maize	NTS	y = 17.196/(1 + 164.611e^(−0.100x)^)	0.9880 **	51.08 b	8.60 b	0.43 a	0.29 a	60.05 b
		SRS	y = 22.163/(1 + 82.288e^(−0.081x)^)	0.9946 **	54.38 a	11.08 a	0.45 a	0.30 a	73.99 a

The mean values of the different treatments in the two cropping systems and two years are shown. Different letters indicate statistically significant differences at the *p* < 0.05 level (ANOVA and Duncan’s multiple range test; n = 3). **: significant at *p* < 0.01.Tmax: maximum dry matter accumulation rate time; Wmax: maximum dry matter accumulation rate of growth; Gmax: maximum dry matter accumulation rate; Gmean: the mean dry matter accumulation rate; D: active dry matter accumulation phase. NTS: no-till with single row planting; SRS: strip deep rotary combined with staggered planting.

**Table 2 plants-12-02000-t002:** Grain yield and yield components by different management practices in 2015 and 2016.

Year	Cropping System	Treatment	Harvest Ears Number (×10^4^ ha^−1^)	Grain Number Per Ear	1000-Grain Weight	Yield (kg ha^−1^)
2015	Spring maize	NTS	7.73 b	460.11 c	315.09 d	11087.54 b
		SRS	7.79 b	468.31 bc	327.82 c	11909.53 a
	Summer maize	NTS	8.44 a	472.96 b	270.88 f	10810.16 b
		SRS	8.29 a	489.18 a	282.76 e	11536.05 a
2016	Spring maize	NTS	5.71 d	362.26 g	372.14 a	7634.94 d
		SRS	7.05 c	389.08 f	349.91 b	9577.65 c
	Summer maize	NTS	7.64 b	414.28 e	251.58 g	7961.85 d
		SRS	7.68 b	444.96 d	267.85 f	9149.02 c
Source of variation						
Year (Y)			***	***	***	**
Cropping system (C)			***	***	**	ns
Management practices (M)			*	***	*	***
Y × M			**	**	**	**
C × M			**	ns	**	ns
Y × C			**	***	***	ns
Y × C × M			*	ns	**	ns

The mean values of the different treatments in the two planting seasons and two years are shown. Different letters indicate statistically significant differences at the *p* <0.05 level (ANOVA and Duncan’s multiple range test; n = 3). *: significant at *p* < 0.05; **: significant at *p* < 0.01; ***: significant at *p* < 0.001; ns: not significant, *p* > 0.05. NTS: no-till with single row planting; SRS: strip deep rotary combined with staggered planting.

**Table 3 plants-12-02000-t003:** The phenological stages of maize under two cropping systems in 2015–2016.

Year	Maize	Sowing Date (M-D)	Emergence Date (M-D)	V6 Date (M-D)	V11 Date (M-D)	Anthesis Date (M-D)	Harvesting Date (M-D)
2015	Spring maize	17 May	24 May	2 June	5 July	16 July	18 September
	Summer maize	28 June	3 July	23 July	7 August	19 August	23 October
2016	Spring maize	19 May	25 May	23 June	7 July	19 July	22 September
	Summer maize	29 June	5 July	29 July	10 August	19 August	20 October

## Data Availability

Electronic data associated with this study can be requested from Sun, X.F. (sunxuefang@qau.edu.cn).

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
