# Peer review of "Integrated Management Practices for Canopy–Topsoil Improves the Grain Yield of Maize with High Planting Density"

_plants, 2023, doi:10.3390/plants12102000_

Round 1
Reviewer 1 Report
The manuscript is well written an is aligned with the journal's scope. The results provides new insights about how maize yield with high plating density can be improved. I have no major concern about this work,nevertheless, I strongly advise that the authors change the wordings in particularly in the material and method section. Most of the provided materials have high similarities with the previous published work. While in some cases it is inevitable, the authors need to modify their text in the M&M section to avoid potential conflict in the similarity checks.
Additionally, there are a couple of suggestions and errors in the manuscript that I highlighted/commented in an annotated PDF.

Reviewer 2 Report
Maize is one of the most popular crops worldwide. The article is of interest for solving the problems of maize cultivation, increasing the yield of this important agricultural crop. The authors also develop modern approaches to sowing plants and tillage in order to increase productivity. This is a hot topic given the growing population of the planet, the changing climate and the lack of food in many countries of the world. The authors are presented with a number of questions and comments to improve the work.
1. Figure 1 shows two maize cultivation systems. Which one is the control of the experiment?
2. In fig. 2 in 2016 shows a significant peak in precipitation. Could rains influence the results of the experiment? Why did the authors choose two years of study rather than four?
3. Line 221 number 243.04% is not a mistake?
4. Line 276 -16.84% is not a mistake?
5. Line 322 the authors write that the yield was affected by the year. How can the year affect the yield? Planting method, humidity yes, but not year.
6. Below, the authors write that in one year there was a difference in the number of maize cobs between the methods, but not in another year. That is, other factors, besides the landing method, influenced this process.
7. Excessive citation of articles by authors from one country. It is recommended to expand the range of cited literature sources from other countries.
Round 2
Reviewer 1 Report
The authors have adequately responded to my concerns. The manuscript can be accepted for publication.
Reviewer 2 Report
The authors corrected some comments. Unfortunately, the authors did not answer a number of questions, so it is difficult to decide if they agree with the comments or not. In general, the article has improved significantly and can be approved for publication.